# Impact of Climate on the Carbon Sink Capacity of Ecological Spaces: A Case Study from the Beijing–Tianjin–Hebei Urban Agglomeration

**Xinyan Wang, Kaiping Wang, Yunlu Zhang \*, Jingran Gao and Yiming Xiong**

School of Landscape Architecture, Beijing Forestry University, Beijing 100107, China; wangsia@bjfu.edu.cn (X.W.); wangkaiping0224@bjfu.edu.cn (K.W.); annie_frank@bjfu.edu.cn (J.G.); itreexym01@163.com (Y.X.)

\* Correspondence: zhangyunlu@bjfu.edu.cn

**Abstract:** Climate plays a significant role in shaping ecosystem-level carbon sinks. Research on the mechanisms of climate impacts on carbon sinks can contribute to the achievement of carbon neutrality. Investigating the mechanisms by which climate impact on carbon sinks in ecological spaces in the Beijing–Tianjin–Hebei region, one of the most important urban clusters in China, is of great significance. This study employed spatial autocorrelation and econometric models to explore how various climatic factors impact net primary productivity (which is used to represent carbon sink capacity) on a spatial scale. We found an increasing trend in NPP across the Beijing–Tianjin–Hebei urban agglomeration from 2000 to 2020, with marked spatial clustering. Climatic factors exhibited the best fit with the spatial Durbin model, except for average annual precipitation. The remaining factors had significant effects on NPP, showing spatial spillover effects. Results also showed that the average annual temperature, evaporation, and relative humidity had positive impacts on NPP at a local scale but adverse effects at a regional scale. Average annual sunshine duration and the ground temperature had negative effects on NPP locally but promoted effects regionally. Furthermore, the average annual wind speed negatively impacted both local- and regional-scale NPP. This research provides insights into how climate affects carbon sinks on a small spatial scale, offering important references for making policy decisions and improving the accuracy of carbon cycling simulations.

**Keywords:** climate; carbon sinks; spatial effects; Beijing–Tianjin–Hebei urban agglomeration; econometric models

## 1. Introduction

The dual carbon strategy is gaining significant attention as a key approach to global climate governance, aiming to protect the environment and promote a shared future for all humanity [1]. For instance, in 2020, China pledged to reach peak carbon emissions by 2030 and to achieve carbon neutrality by 2060 [2]. Therefore, strengthening regional resilience to climate uncertainties and threats is the key to addressing climate change and sustainable regional development [3]. Enhancing carbon sinks is an important way to mitigate and adapt to climate change and achieve the goal of the dual carbon strategy [4]. The concept of the "three zones" [5] emphasizes the sustainable development and conservation of regional ecosystems through ecological spaces such as grasslands, forests, and wetlands [6], which are important areas for carbon sinks. Therefore, eco-spatial carbon sinks are an important target for our attention [7]. Net Primary Productivity (NPP), or the production of plant biomass, is equal to all of the carbon taken up by the vegetation through photosynthesis (called Gross Primary Production or GPP) minus the energy lost to respiration. NPP = GPP − respiration, which is used to represent carbon sink capacity.

Studies have shown that socio-economic factors (e.g., economic development and population size) [8], natural factors (e.g., $CO_2$ concentration in the atmosphere, forest structure and density [9,10], and plant-available nitrogen [11]), and anthropogenic disturbances (e.g.,

land use changes) are crucial in determining carbon sinks [12]. Additionally, climate can directly affect carbon sinks at various scales, making it a significant factor in driving carbon sinks [13]. However, research exploring the relationship between climate and carbon sinks tends to focus on a macroscale, such as a global scale, where a temperature rise is often linked to a decrease in NEP (Net Ecosystem Productivity) [14]. However, the temperature fluctuates widely across ecosystems at different scales, with different impacts on carbon sinks. For example, temperature fluctuations have been shown to adversely affect carbon sinks in high-altitude ecosystems [15]. At the same time, water availability is highly sensitive to interannual carbon balance in semi-arid and subtropical ecosystems [16]. These findings highlight the variable relationships between climatic factors and carbon sinks across the ecosystems [17]. However, more research is needed to explore relationships between climate, particularly local and regional, and carbon sinks [18].

Terrestrial ecosystems in China experience a small interannual variability in carbon balance [19]. However, different regions experience varying sensitivity to precipitation and temperature and carbon sink response to climate change [20]. Despite the high variability in precipitation and temperature across the ecosystems, the available literature suggests that only these two macroscale variables have been used to assess climate impact on carbon sinks [21]. More importantly, the role of local-scale climatic variables, such as wind speed, sunshine duration, relative humidity, and evaporation, in determining the ecosystem's carbon sink is often overlooked [22].

In recent years, as urban agglomerations have demonstrated the advantages of synergistic development and governance in environmental management globally, more carbon-sink-related studies have been conducted at the regional scale [23]. Studies have mainly adopted methods such as trend analysis, superposition analysis, and correlation coefficients, mostly focusing on the perspective of dynamic change analysis of driving factors, with more emphasis on time series analysis but less research on the correlation aspect of static spatial differentiation [24,25]. The spatial Durbin model can quantitatively detect the explanatory power of climate factors on NPP and spatial interaction, which can better address the limitations in the above studies [26].

The Beijing–Tianjin–Hebei (in the following, the Beijing–Tianjin–Hebei is abbreviated as B-T-H) urban agglomeration is a strategic urban region driving rapid socio-economic development in China [27]. In 2020, the region's energy consumption reached 480 million tons of standard coal, accounting for 9.6% of the national total energy consumption in China [28]. Increased energy consumption also enhances carbon emissions [29]. To mitigate carbon emissions, the "Outline of the B-T-H Coordinated Development Plan" [30] highlights the need for carbon sink enhancement measures, and the region should lead the way in coordinated development and reform, drive innovation-driven economic growth nationwide, and serve as a demonstration zone for ecological restoration [31]. Given the sound basis of synergistic development in the B-T-H region, it is an important pilot demonstration site to explore the mechanisms of climate impacts on carbon sinks [32].

In view of the prior studies, this paper drew on existing research, selected seven climate and phenological factors, and used spatial econometric models to complement the mechanisms of climate influence on carbon sinks in ecological spaces in the B-T-H urban agglomeration region [33]. Using 20-year data, this paper aimed for the following: (1) to investigate the spatial and temporal patterns of carbon sinks in ecological spaces in the B-T-H region; (2) to understand the mechanisms by which climate factors drive carbon sinks at small scales using spatial econometric models; (3) to estimate the spatial spillover effects of carbon sinks using a spatial Durbin model parameterization; and (4) to decompose the spatial spillover effects of climate drivers and explore the mechanism of action for the region and neighboring regions. The target of the study is to provide more scientific guidance for increasing the development of carbon sink policies in the B-T-H region and to increase the accuracy of local- and regional-scale carbon cycle simulations.

## 2. Materials and Methods

### 2.1. Study Area

The B-T-H urban agglomeration is located in Northern China and consists of two municipalities (Beijing and Tianjin) and 11 prefecture-level cities (Figure 1). The region experiences a temperate continental monsoon climate characterized by hot and humid summers and cold and dry winters [34]. The average annual temperature ranges from 11.5 to 12.5 °C, and the annual precipitation ranges from 531 to 644 mm. The ecological space across the B-T-H region comprised three important ecological functional areas: the water conservation area in the northern B-T-H region, the water conservation and soil conservation area in the Taihang Mountain area, and the wind and sand fixation area in the Hunshandake Sandy Land [35]. According to the third national land and space survey, the region occupied 218,000 square kilometers, with cultivated land covering 64,572.8 square Km and construction land covering 27,488.1 square Km. More than 44% of the total area was covered by ecological space, providing vast prospects for green economic development.

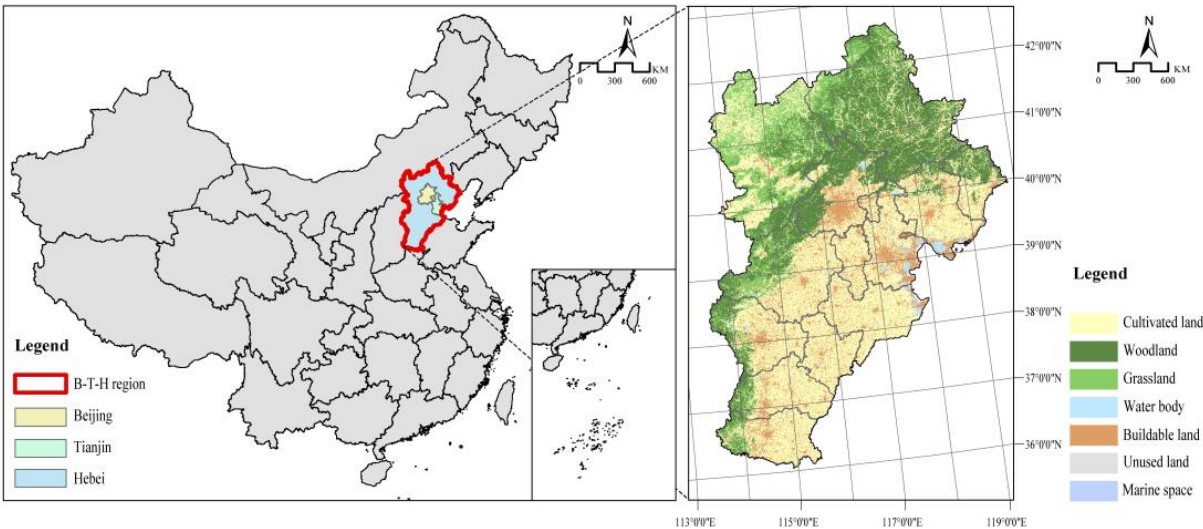

**Figure 1.** Location map of the study area showing the B-T-H region.

### 2.2. Variable Selection and Data Sources

#### 2.2.1. Variable Selection

The impact of climate on carbon sinks in the B-T-H ecological space was studied from 2000 to 2020 at 10-year intervals by measuring NPP. NPP is the rate plants absorb carbon dioxide from the atmosphere and convert it into carbohydrates through photosynthesis. As the rate of net plant biomass grows, NPP can be considered as the net carbon uptake by an ecosystem. Therefore, higher NPP values generally indicate a stronger carbon sink capacity of an ecosystem [36].

Seven climate-related variables, average temperature (Tem), annual precipitation (Pre), annual evaporation (Evp), annual average wind speed (Win), annual sunshine duration (Ssd), annual average relative humidity (Rhu), and annual ground temperature (Gst), were selected as explanatory variables. Considering the spatial autocorrelation and data processability, the predictor and response variables were acquired by dividing the study region into grids (20 km × 20 km), obtaining 508 raster cells. We divided the region into grids to accurately and completely represent the unevenly distributed ecological spaces at the county level and align with the coordinated implementation of the developmental plan of the B-T-H region. The grids were then identified to establish a geographical weight matrix.

### 2.2.2. Data Sources

NPP data were obtained from the Resource and Environment Science and Data Center (https://www.resdc.cn/) (accessed on 1 May 2023). NPP was calculated using the light energy utilization model, the GLO_PEM (Global Primary Energy Model).

Climate data were acquired on the annual spatially interpolated dataset of meteorological elements from the Resource and Environment Science and Data Center (https://www.resdc.cn/DOI/DOI.aspx?DOIID=96) (accessed on 1 May 2023). Land use data were obtained from Wuhan University with a 30-meter resolution. Socio-economic data were sourced from the "Beijing Statistical Yearbook (beijing.gov.cn)" (accessed on 1 May 2023), "Tianjin Statistical Yearbook (tianjin.gov.cn)" (accessed on 1 May 2023), and "Hebei Statistical Yearbook (hebei.gov.cn)" (accessed on 1 June 2023). Statistical and computational analyses were conducted using ArcGIS 10.8 and Stata. The spatial weight matrix was performed using Geoda software (http://geodacenter.github.io/) (accessed on 1 May 2023).

### 2.2.3. Statistical Methods

We explored the spatial attributes of carbon sinks across ecological spaces in the B-T-H agglomeration using the most common spatial analyses, the Global Moran's I, Local Moran's I, and the spatial econometric model.

1.   Moran's I

(1)   Global Moran's I

Empirical evidence suggests that climate factors exhibit both spatial dependence and heterogeneity. Therefore, global Moran's I was employed to examine the spatial attributes of carbon sinks across ecological spaces. Global Moran's I is the most widely used measure of spatial autocorrelation in ecological and environmental studies, which assesses the degree of clustering among geographic features. Moran's I calculates spatial autocorrelation by comparing the correlation between the attribute values of each geographic unit and those of its neighboring units. Moran's I was calculated as follows:

$$\text{Moran's I} = \frac{N}{s_0} \frac{\sum_{i=1}^{N} \sum_{j=1}^{N} \omega_{ij}(y_i - \overline{y})(y_i - \overline{y})}{\sum_{i}^{N}(y_i - \overline{y})^2}$$

In Moran's I, $s_0$ was calculated as follows:

$$s_0 = \sum_{i=1}^{N} \sum_{j=1}^{N} \omega_{ij}$$

Moran's I statistics range from −1 to 1, where positive values indicate positive spatial autocorrelation or clustering (i.e., neighboring units are similar), suggesting spatial dependency. In contrast, negative values indicate negative spatial autocorrelation or spatial dispersion (i.e., neighboring values are dissimilar), suggesting inverse spatial dependency. An autocorrelation value of 0 implies a random distribution in space.

(2)   Local Moran's I

The local Moran's I is a local autocorrelation statistic based on global Moran's I and provides the contribution of each spatial unit's observation. The local Moran's I statistic was calculated as follows:

$$\text{Local Moran's I} = \frac{y_i - \overline{y}}{S_i^2} \sum_{j=1, j \neq 1}^{N} \omega_{ij}(y_i - \overline{y})$$

where $S_i^2$ is defined as follows:

$$s_i^2 = \frac{\sum_{j=1, j \neq 1}^{N} \omega_{ij}}{N - 1} - \bar{y}^2$$

where $N$ represents the number of spatial grids (observations), $s_0$ represents the sum of all elements in the spatial weight matrix, $y_i$ represents the observed value of variable y in spatial unit I, and $\omega_{ij}$ represents the element in the spatial weight matrix.

2. Spatial econometrics models

(1) The Ordinary Least Squares (OLS) model

The OLS is a statistical method used to estimate the relationship between a dependent variable and one or more independent variables. It is widely used in regression analysis. The OLS models aim to minimize the sum of squared differences between observed and predicted values. The formula for the OLS model can be presented as

$$Y = \beta_0 + \beta_1 X_1 + \beta_2 X_2 + \ldots + \beta_n X_n + \varepsilon$$

where Y is the dependent variable; $X_1$, $X_2$, ..., $X_n$ are the independent variables; $\beta_0$, $\beta_1$, $\beta_2$, ..., $\beta_n$ are the coefficients representing the relationship between the dependent and independent variables; and $\varepsilon$ is the error term representing the unexplained variation in the dependent variable.

(2) Spatial weight matrix:

A spatial weight matrix [508, 508] was constructed using the First Law of Geography, which states that the relationship between geographic regions weakened as the geographic distance increased. The Geoda software was used to construct a binary spatial weight matrix, which was then converted to a standardized spatial weight matrix with $508 \times 508$ grids using Stata. The equation for spatial econometrics modeling is as follows:

$$w = \begin{bmatrix} w_{11} w_{12} \cdots w_{1n} \\ w_{21} w_{22} \cdots w_{2n} \\ \vdots \quad \vdots \quad\quad \vdots \\ w_{n1} w_{n2} \cdots w_{nn} \end{bmatrix}$$

Spatial econometric models are statistical models used to analyze spatial correlation and dependence. They extend traditional econometric models by incorporating spatial factors to capture the mutual influence and dependency between neighboring areas in geographic space. Specifically, spatial econometric models introduce a spatial weight matrix, which measures spatial correlation and spatial adjacency among different areas, reflecting the degree of interconnectivity in geographic space. The matrix includes endogenous interaction (WY) and exogenous interaction (WX). The most commonly used spatial econometric models include the Spatial Lag Models (SLM) and Spatial Error Models (SEM).

The Spatial Durbin Model (SDM) simultaneously incorporates *WY* and *WX* interaction effects and is expressed as follows:

$$Y = \rho WY + X\beta + WX\theta + \varepsilon, \varepsilon \sim N\left(0, \delta^2\right)$$

where $Y$ represents the dependent variable, $W$ is the spatial weight matrix used to capture the spatial correlation between the sample units, $X$ is a matrix of independent variables, $\beta$ is a coefficient vector of the independent variables, $\theta$ is a coefficient for the exogenous interaction effect, $WX$ is a matrix of spatially lagged independent variables, and $\varepsilon$ is an error term. Assuming that $\varepsilon$ follows a multivariate normal distribution, with zero mean and a constant scalar diagonal variance–covariance matrix, $\delta^2$. When $\theta = 0$, it corresponds

to the SLM. When $\theta = -\rho\beta$, it corresponds to the SEM. When the error term of the model exhibits spatial correlation, it is known as the SLM and expressed as follows:

$$Y = \rho WY + X\beta + \varepsilon, \varepsilon \sim N\left(0, \delta^2\right)$$

When the spatial dependency among the dependent variables leads to spatial correlation in the model, it is referred to as the Spatial Error Model, also known as the SEM, and is expressed as follows:

$$Y = X\beta + \lambda W_u + \varepsilon, \varepsilon \sim N\left(0, \delta^2\right)$$

where $u$ is the random error vector, and $\lambda$ is the spatial correlation coefficient among the regression residuals.

## 3. Results

### 3.1. Spatial and Temporal Variability of Ecological Spatial Carbon Sinks in B-T-H Region

The study first calculated trends in NPP for 2000, 2010, and 2020 (Figure 2). From 2000 to 2020, the total NPP in the ecological spaces across urban agglomeration initially showed an increasing trend. The total NPP was 131,264 gC in 2000 and 155,274 gC in 2010, while it was 195,821 gC in 2020. Combining NPP distribution and spatial distribution pattern with B-T-H topography (Figure 3a) and B-T-H ecosystem support area planning (Figure 3b), NPP was lower in northwest regions but higher in the north and west regions. Correspondingly, the NPP value of the northwestern dam steppe in which the Bashang Plateau Ecological Protection Zone in the B-T-H region is located is relatively low, while the NPP of the Yanshan-Taipei Mountains Ecological Protection Zone is high [37].

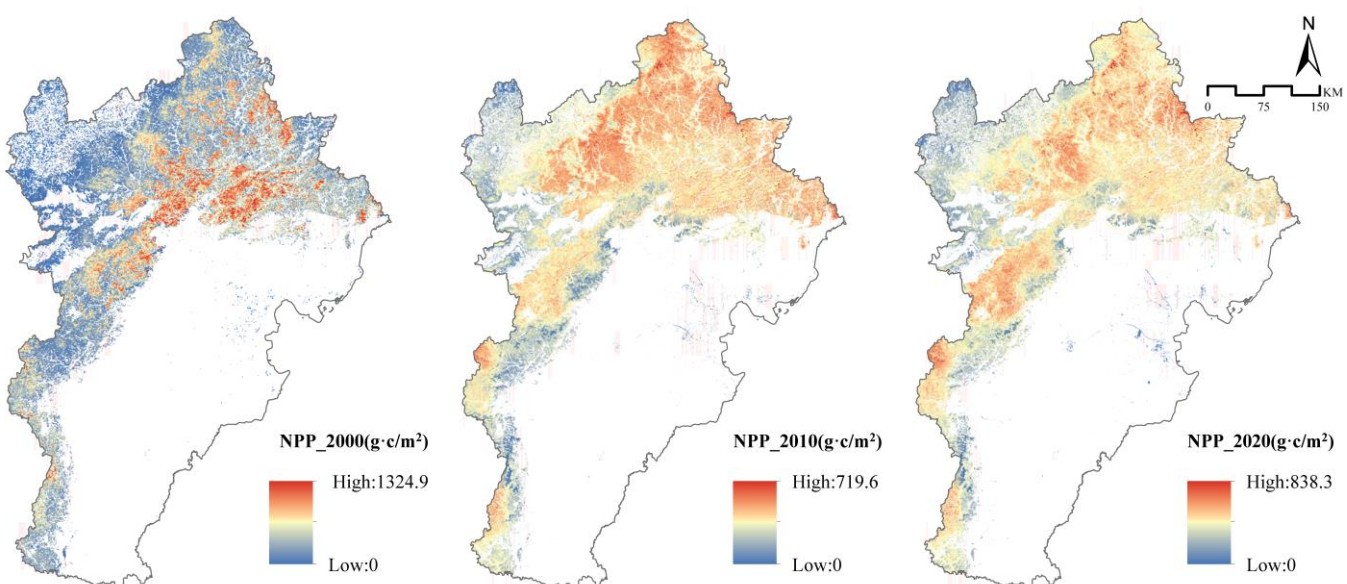

**Figure 2.** Net primary productivity of ecological spaces in the B-T-H region in 2000, 2010, and 2020.

Before applying spatial econometric models, we used global and local Moran's I statistics to analyze the spatial autocorrelation of NPP in ecological spaces. The Moran's I values were 0.555, 0.335, and 0.798, respectively, for 2000, 2010, and 2020, and the corresponding z-values were 22.276, 13.884, and 32.124 (Table 1). The Moran's I values remained positive throughout the observation period, indicating that the spatial distribution of NPP exhibited either a high–high (H-H) or low–low (L-L) pattern with significant spatial autocorrelation.

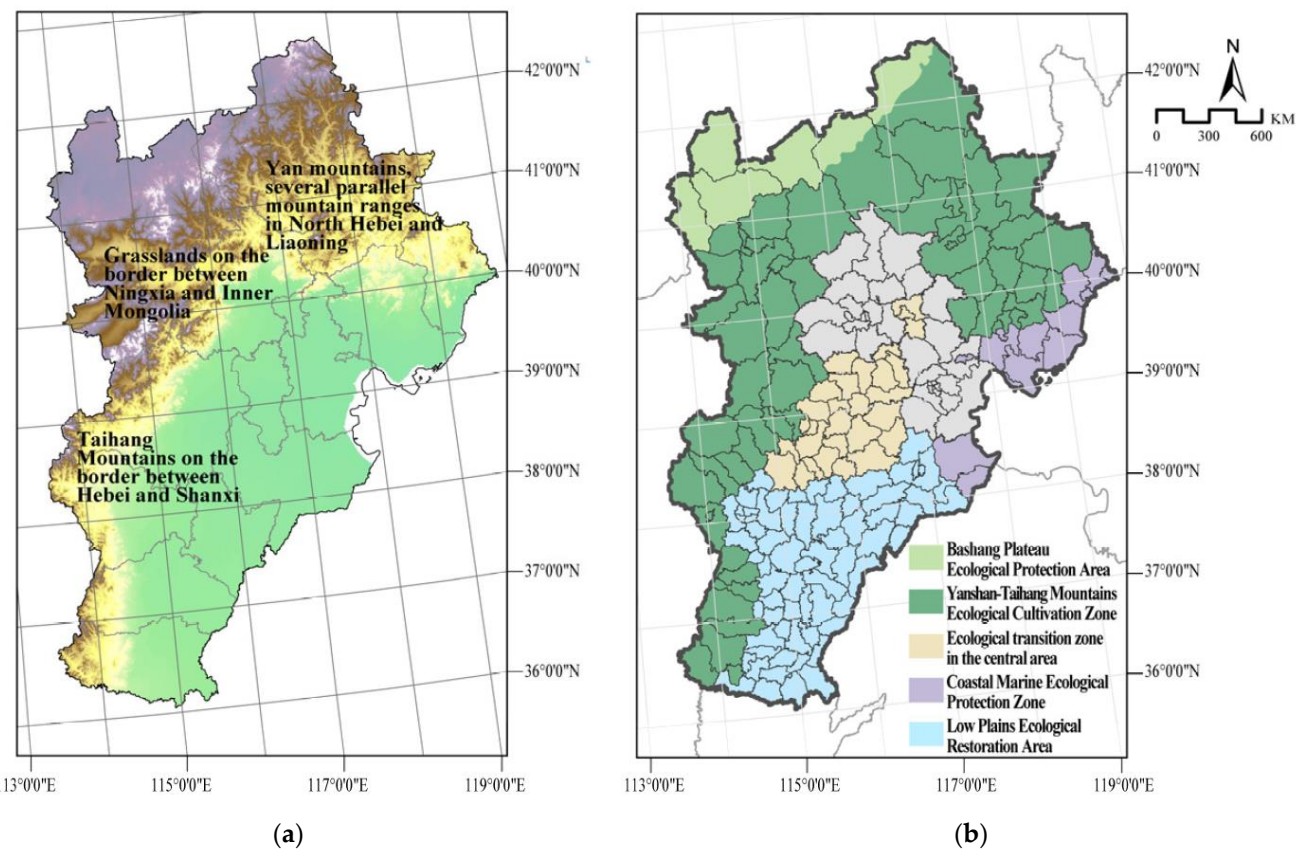

**Figure 3.** (**a**) Topographic map of B-T-H region (**b**) B-T-H ecosystem support area planning.

**Table 1.** Global Moran's index for eco-spatial NPP in the Beijing–Tianjin–Hebei region.

| Variables | I | E(I) | sd(I) | z | *p*-Value |
|-----------|-------|--------|-------|--------|-----------|
| **NPP2000** | 0.555 | −0.002 | 0.025 | 22.276 | 0.000 |
| **NPP_2010** | 0.335 | −0.002 | 0.024 | 13.884 | 0.000 |
| **NPP2020** | 0.798 | −0.002 | 0.025 | 32.124 | 0.000 |

The Moran scatter plots for the years 2000, 2010, and 2020 also showed that NPP exhibited high–high (H–H) or low–low (L–L) spatial distribution patterns. The spatial autocorrelation in NPP observed from Moran's I suggested a need for further analysis using spatial econometric models (Figure 4).

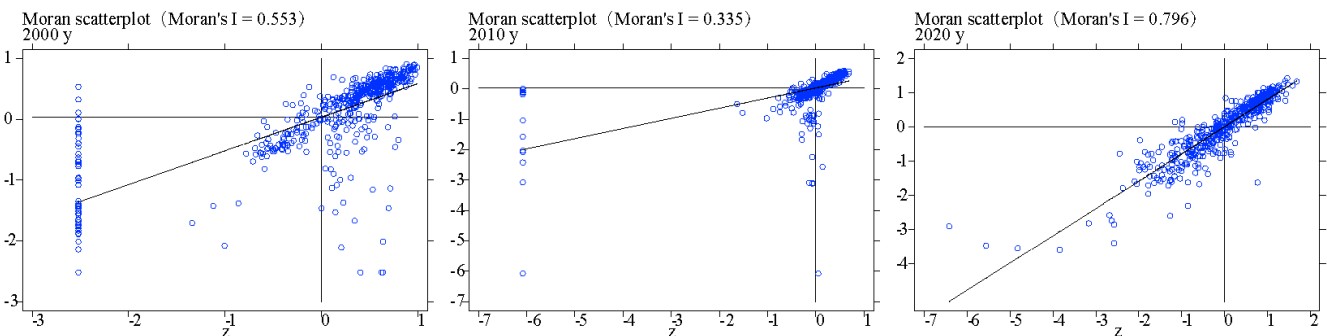

**Figure 4.** LISA cluster diagram of ecological space NPP in B-T-H region during 2000–2020.

*3.2. Estimation of Spatial Measurement Models*

3.2.1. Ordinary Least Squares Regression and Selection Tests for Spatial Models

Ordinary Least Squares regression analysis was employed to relate average annual NPP with various climatic factors. The R-squared value (>0.5) indicated a relatively good fit of the models (Table 2). As the use of OLS regression (Table 2) alone to estimate the effects of explanatory variables on NPP may create certain biases, further tests will be conducted applying spatial models.

**Table 2.** OLS regression model test results.

|  | ols_ind | ols_time | ols_spatiotemporal | ols_random |
|---|---|---|---|---|
| **Tem** | −0.207 * | −0.020 | −0.436 *** | −0.214 *** |
| **Pre** | −0.097 | 1.168 *** | −0.875 *** | 1.525 *** |
| **Evp** | −1.338 *** | 0.625 ** | 1.928 *** | −1.199 *** |
| **Win** | −0.836 *** | −0.832 *** | −1.138 *** | −0.641 *** |
| **Ssd** | 2.382 *** | −0.518 * | 0.515 | 0.869 *** |
| **Rhu** | −0.096 *** | −0.006 | −0.042 * | −0.034 *** |
| **Gst** | 0.396 *** | −0.127 ** | 0.078 | 0.161 *** |
| **_cons** | 2.144 *** | 1.051 *** | −0.044 | 1.424 *** |
| **$R^2$** | 0.516 | 0.521 | 0.564 | 0.477 |
| **N** | 1524.000 | 1524.000 | 1524.000 | 1524.000 |

*, **, and ***, respectively, indicate the significance levels at 10%, 5%, and 1%.

According to the above Moran's I index calculation results, there is an obvious positive spatial spillover effect of the ecological space NPP in Beijing–Tianjin–Hebei, and considering the effect of the interaction term, the SDM model, SLM model, and SEM model are chosen to analyze the effect of different climatic factors on NPP. However, it is necessary to choose the appropriate measurement model according to the test and determination rules, and the test results are shown in Table 3.

**Table 3.** LM, LR, and Hausman testing.

| Methods | _cons | *p* | Methods | _cons | *p* |
|---|---|---|---|---|---|
| **LM–spatial lag** | 97.890 | 0.000 | Wald–spatial lag | 254.78 | 0.000 |
| **Robust LM–spatial lag** | 13.702 | 0.000 | LR–spatial lag | 289.62 | 0.000 |
| **LM–spatial error** | 321.628 | 0.000 | Wald–spatial error | 138.98 | 0.000 |
| **Robust LM–spatial error** | 237.441 | 0.000 | LR–spatial error | 380.27 | 0.000 |

First, the Lagrange Multiplier (LM) and Robust LM tests (Table 3) were combined to determine the form of spatial correlation (whether it exists in the error term, SEM, or the lag term, SLM). Second, the Wald test and Likelihood Ratio (LR) test were used to determine whether the Spatial Durbin Model (SDM) could be simplified into the Spatial Error Model (SEM) and Spatial Lag Model (SLM). If the original hypothesis is rejected at the same time, then SDM is the best-fit model; if the results show the rejection of the original hypothesis of the Wald test and the Robust LM lag value is significant, then the SDM model cannot be optimized as an SLM model; if the results show the rejection of the original hypothesis of the LR test and the Robust LM lag value is significant, then the SDM model cannot be optimized as an SEM model. According to the test results in the table, all the statistics in the LM test were significant, indicating that both the SLM and SEM models were applicable. Subsequently, the Wald test and LR test comparing the SEM model and the SLM model rejected the original hypothesis and passed the 1% significance test, so the SDM model could not be optimized for the SLM model and the SEM model. In the end, the SDM was used for regression analysis.

### 3.2.2. Spatial Durbin Model Results

Regression results were compared using SDM random effect, spatial fixed effect, time fixed effect, and spatiotemporal fixed effect. The spatiotemporal fixed effects produced more significant results. Integrating with the Hausman test, the fixed effect model was chosen, as it exhibited lower endogeneity. Finally, a spatiotemporal fixed effect model was selected for analysis, and the results are presented in Table 4.

**Table 4.** Estimated results of SDM.

| | SDM_ind | SDM_time | SDM_spatiotemporal | SDM_random |
|---|---|---|---|---|
| **Tem** | 1.217 *** | 0.728 ** | 1.237 *** | 0.523 |
| | (3.67) | (2.16) | (3.74) | (1.57) |
| **Pre** | −0.238 | −0.120 | −0.155 | −0.245 |
| | (−0.50) | (−0.27) | (−0.33) | (−0.56) |
| **Evp** | 8.543 *** | 8.778 *** | 8.659 *** | 8.545 *** |
| | (11.07) | (12.42) | (11.25) | (12.18) |
| **Win** | −0.725 ** | −1.700 *** | −0.692 * | −1.602 *** |
| | (−2.04) | (−5.33) | (−1.96) | (−5.03) |
| **Ssd** | −6.794 *** | −6.569 *** | −6.840 *** | −6.597 *** |
| | (−8.36) | (−8.72) | (−8.44) | (−8.79) |
| **Rhu** | 0.113 ** | 0.078 ** | 0.098 ** | 0.105 *** |
| | (2.53) | (2.04) | (2.19) | (2.80) |
| **Gst** | −1.777 *** | −1.360 *** | −1.806 *** | −1.124 *** |
| | (−5.07) | (−3.84) | (−5.18) | (−3.21) |
| **_cons** | | | | 0.498 *** |
| | | | | (4.28) |
| **Wx** | | | | |
| **Wx Tem** | −1.561 *** | −0.803 ** | −1.733 *** | −0.612 * |
| | (−4.32) | (−2.29) | (−4.78) | (−1.77) |
| **Wx Pre** | −0.080 | 0.612 | −0.479 | 0.784 * |
| | (−0.15) | (1.28) | (−0.89) | (1.65) |
| **Wx Evp** | −9.396 *** | −9.257 *** | −8.840 *** | −9.240 *** |
| | (−11.42) | (−12.29) | (−10.21) | (−12.62) |
| **Wx Win** | 0.174 | 1.375 *** | −0.001 | 1.296 *** |
| | (0.45) | (4.02) | (−0.00) | (3.82) |
| **Wx Ssd** | 8.992 *** | 7.078 *** | 8.950 *** | 7.279 *** |
| | (10.04) | (8.88) | (9.89) | (9.23) |
| **Wx Rhu** | −0.196 *** | −0.092 ** | −0.169 *** | −0.127 *** |
| | (−3.86) | (−2.25) | (−3.29) | (−3.17) |
| **Wx Gst** | 2.044 *** | 1.417 *** | 2.025 *** | 1.205 *** |
| | (5.55) | (3.83) | (5.51) | (3.32) |
| **Spatial rho** | 0.543 *** | 0.587 *** | 0.523 *** | 0.580 *** |
| | (20.41) | (22.01) | (18.91) | (22.40) |
| **Variance sigma2_e** | 0.285 *** | 0.470 *** | 0.283 *** | 0.440 *** |
| | (27.18) | (27.27) | (27.23) | (21.94) |
| **lgt_theta** | | | | 2.566 *** |
| | | | | (4.59) |

*, **, and ***, respectively, indicate the significance levels at 10%, 5%, and 1%.

Results (Table 5) showed that annual average temperature, evaporation, and relative humidity had significant positive effects on NPP ($p < 0.01$). In contrast, annual average sunshine duration and ground temperature had significant negative effects ($p < 0.01$). Additionally, annual average wind speed showed a significant negative effect on spatial carbon sequestration ($p < 0.1$). However, annual average precipitation showed mixed results across multiple models, with no consistent effects.

**Table 5.** Results showing direct and indirect effects of climate variables on NPP.

|  | **Direct** | **Indirect** | **Total** |
|---|---|---|---|
| Tem | 1.125 *** | −2.167 *** | −1.042 *** |
|  | (3.52) | (−5.20) | (−4.36) |
| Pre | −0.237 | −1.049 | −1.286 *** |
|  | (−0.55) | (−1.62) | (−2.79) |
| Evp | 8.232 *** | −8.547 *** | −0.315 |
|  | (11.89) | (−8.30) | (−0.43) |
| Win | −0.741 ** | −0.709 * | −1.450 *** |
|  | (−2.27) | (−1.65) | (−7.00) |
| Ssd | −6.268 *** | 10.629 *** | 4.361 *** |
|  | (−8.61) | (9.63) | (5.30) |
| Rhu | 0.085 ** | −0.238 *** | −0.153 *** |
|  | (2.00) | (−4.07) | (−4.10) |
| Gst | −1.696 *** | 2.154 *** | 0.458 *** |
|  | (−4.94) | (5.64) | (4.02) |
| r2 |  |  | 0.199 |
| ll |  |  | −1200.000 |
| aic |  |  | 2540.307 |
| bic |  |  | 2700.179 |
| N |  |  | 1524.000 |

*, **, and ***, respectively, indicate the significance levels at 10%, 5%, and 1%.

### 3.2.3. Analysis of Direct and Spatial Spillover Effects

The direct effect coefficients for annual average temperature, evaporation, and relative humidity were positively significant ($p < 0.01$; Table 6). Conversely, their indirect effect coefficients were negatively significant ($p < 0.01$). These results indicate that annual average temperature, evaporation, and relative humidity positively impact local-scale NPP while restricting carbon sequestration in neighboring areas. The direct effect of annual average sunshine duration and ground temperature on NPP was negative, while their indirect effects were positive. Increases in local annual average sunshine duration and higher ground temperature locally diminish NPP while promoting it in neighboring areas. The direct and indirect effect coefficients of annual average wind speed were −0.741 and −0.709, respectively ($p < 0.05$), exhibiting diminishing effects on NPP at local and adjacent areas.

### 3.2.4. Robustness Test

The relationships between the carbon sequestration capacity of ecological spaces at a spatial scale and climate variables partly conformed to the SEM and SLM (Table 6). However, both fits were not as good as the SDM, confirming the robustness of the Wald test and LR test in Section 3.2.1 and justifying the use of SDM.

**Table 6.** Results from SEM and SLM models.

| | SEM_ind | SEM_time | SEM_both | | SLM_ind | SLM_time | SLM_both | SLM_random |
|---|---|---|---|---|---|---|---|---|
| **Main** | | | | **Main** | | | | |
| Tem | −0.462 *** (−3.31) | 0.142 (1.26) | −0.165 (−1.25) | x1 | −0.258 *** (−3.06) | −0.003 (−0.06) | −0.415 *** (−4.96) | −0.132 *** (−3.11) |
| Pre | 0.400 (1.11) | 1.054 *** (4.07) | −0.424 (−1.16) | x2 | 0.029 (0.14) | 0.685 *** (5.37) | −0.549 *** (−2.65) | 0.836 *** (6.47) |
| Evp | 0.279 (0.53) | 4.999 *** (7.05) | 6.030 *** (9.24) | x3 | −0.481 ** (−2.24) | 0.688 *** (2.62) | 1.623 *** (5.07) | −0.482 ** (−2.53) |
| Win | −0.718 *** (−3.98) | −1.133 *** (−8.17) | −1.016 *** (−5.44) | x4 | −0.421 *** (−4.54) | −0.528 *** (−8.60) | −0.686 *** (−7.14) | −0.384 *** (−6.54) |
| Ssd | 0.263 (0.40) | −4.292 *** (−6.61) | −4.124 *** (−5.80) | x5 | 1.424 *** (4.38) | −0.552 ** (−2.13) | 0.302 (0.86) | 0.375 * (1.70) |
| Rhu | −0.068 ** (−2.50) | 0.017 (0.93) | 0.024 (0.82) | x6 | −0.067 *** (−4.37) | 0.004 (0.52) | −0.030 * (−1.88) | −0.014 * (−1.69) |
| Gst | 0.546 *** (5.45) | −0.494 *** (−3.79) | −0.268 ** (−2.28) | x7 | 0.232 *** (4.82) | −0.094 * (−1.79) | 0.036 (0.69) | 0.102 ** (2.33) |
| lambda | 0.575 *** (14.31) | 0.673 *** (21.04) | 0.632 *** (20.74) | Rho | 0.484 *** (19.50) | 0.426 *** (16.63) | | (1.06) |
| sigma2_e | 0.363 *** (26.48) | 0.497 *** (26.33) | 0.307 *** (26.69) | sigma2_e | 0.353 *** (27.34) | 0.569 *** (27.23) | (16.85) | (18.27) |
| r2 | 0.414 | 0.088 | 0.044 | r2 | 0.479 | 0.440 | 0.336 *** | 0.552 *** |
| ll | −1400 | −1700 | −1300 | Ll | −1400 | −1800 | (27.40) | (21.95) |
| aic | 2895.524 | 3423.239 | 2665.870 | Aic | 2822.983 | 3524.670 | | 2.737 *** |
| bic | 2943.486 | 3471.201 | 2713.832 | Bic | 2870.945 | 3572.632 | | (4.17) |
| N | 1524.000 | 1524.000 | 1524.000 | N | 1524.000 | 1524.000 | 0.140 | 0.587 |

*, **, and ***, respectively, indicate the significance levels at 10%, 5%, and 1%.

## 4. Discussion

### 4.1. Comparative Analysis of Results

The main objective of this paper is to analyze the influence mechanism of climate factors in spatial carbon sinks in the B-T-H region by incorporating geographic factors into the research [38]. During the study period, the overall trend of spatial carbon sink in ecological spaces in B-T-H was on the rise, and the possible reason for this is the influence brought by the implementation of the Beijing–Tianjin Wind and Sand Source Control Project since 2000. The Beijing–Tianjin Wind and Sand Source Control Project is a control measure for land sanding in the areas around Beijing and Tianjin, which was introduced to consolidate soil and prevent sand as well as to reduce sand and dust weather in Beijing and Tianjin. The project area extends from Damao Banner in Inner Mongolia in the west to Arukolqin Banner in Inner Mongolia in the east, from Daixian County in Shanxi in the south to Dongwuzhumuqin Banner in Inner Mongolia in the north, and it involves 75 counties (banners) in five provinces (districts and municipalities), including Beijing, Tianjin, Hebei, Shanxi, and Inner Mongolia (Figure 5). The first phase of the project took 12 years to implement 7.08 million mu of afforestation, planting 150 million trees, and the second phase of the project, which was launched in 2013, completed 2.139 million mu of afforestation. Over the past 20 years, the project has completed a total of 9.22 million mu of afforestation, and the deserted forested mountains and sandy land "should be green as much as possible", which effectively reversed the expansion of the momentum of sandification. The project has effectively reversed the expansion of desertification and realized the ecological function of carbon sequestration and sink enhancement. With the

implementation of the policy, all five major wind and sand hazard areas in B-T-H have been treated for more than 20 years, significantly increasing the spatial carbon sink capacity [39].

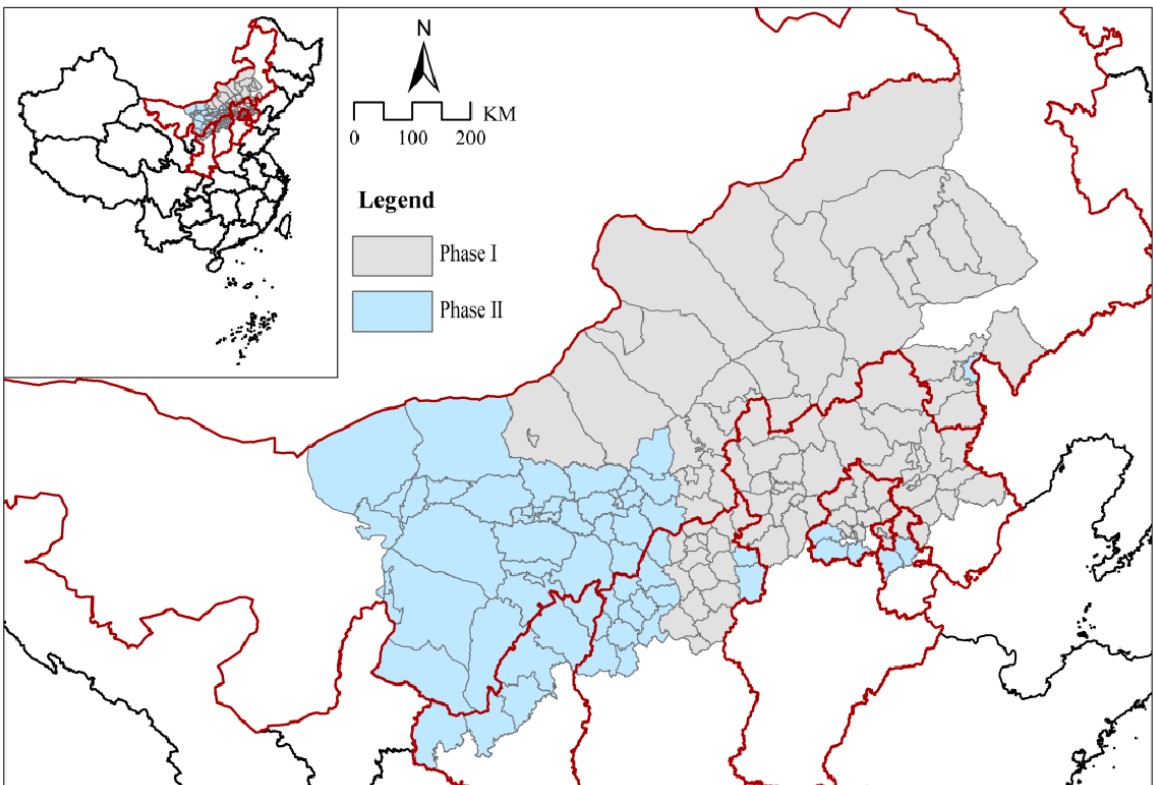

**Figure 5.** Beijing–Tianjin Wind and Sand Source Control Project.

Moran's I statistics showed a significant spatial autocorrelation of carbon sinks in B-T-H, with a high–high and low–low aggregation in spatial distribution. The high NPP areas were located in the Yanshan and Taihang Mountains, the green barriers for the development of B-T-H urban agglomerations. In the context of the integrated development of B-T-H, the "demonstration effect" exists between regions, which can lead the neighboring regions to follow suit and increase the ecological spaces and carbon sinks. Regional synergistic development has been successful in many regions; in Sweden, for example, its climate policy takes into account cross-sectoral equity—i.e., the spatial autonomy of carbon sinks—and multi-level cooperative governance to achieve a win–win situation for both carbon sinks and the economy, which makes Sweden a leading model for climate policy in the EU and other OECD (Organization for Economic Co-operation and Development) countries or regions [40]. The land use of green space is mainly dominated by forests and grasslands, which have long growth cycles, and therefore management is a long-term process [41]. If not taken seriously, huge areas of ecological space can, in turn, become a source of carbon emissions and a barrier to climate mitigation [42].

Compared to the existing literature, this paper not only examined the relationship between climate factor and NPP using traditional modeling methods (e.g., OLS) but also examined its spatial spillover effects using spatial measurement methods and the results of a spatial panel Durbin model of carbon sink with a spatiotemporal effect. Results showed that precipitation had little effect on carbon sinks in ecological spaces in urban clusters; however, many large-scale studies only consider precipitation factors [43,44], which provides limited information on carbon balance and sinks in small regions [45]. The spatial spillover effects of annual average temperature, evaporation, and relative humidity were significant, positively impacting carbon sequestration in local areas but limiting others. One possible reason for these results could be higher temperatures that

cause water to evaporate faster [46], thus increasing relative humidity. In general, climate factors have a consistent impact on carbon sequestration in ecological spaces at different spatial scales [47]. More than rainfall, evapotranspiration and relative humidity at the scale of urban agglomerations can reflect the combined effects of both temperature and the water environment [48]. Sunshine duration also played an important role in carbon sequestration in the B-T-H ecosystem, but the direction of effects varied with the scale. For instance, the annual average sunshine duration and ground temperature showed a negative impact on local-scale carbon sequestration; the same variables increased carbon sequestration in neighboring areas. The scale-dependent effect of sunshine on carbon sequestration is likely due to changes in cloud cover resulting from the interaction between sunshine duration and atmospheric conditions [49], such as an increase in relative humidity that leads to more cloud cover, reducing sunshine duration. Therefore, sunshine duration and relative humidity have contrasting effects on the NPP across ecological spaces. Ground temperature generally affects the carbon sequestration capacity of ecological spaces by influencing vegetation growth. High temperatures inhibit vegetation growth, but the direction of their effect is the opposite [50]. The mechanism of the transition between atmospheric and ground temperatures and its effect on NPP needs further study.

Annual average wind speed had a negative impact on NPP on both the local and regional scales. These findings may be related to the changes in carbon dioxide concentration in the atmosphere caused by higher wind speed as shown in previous studies [51]. As the influence mechanism of carbon sinks is an integrated and complex system, with climate, $CO_2$ concentration, and nitrogen deposition interacting with each other, the role of other driving factors cannot be completely excluded. Therefore, the limitations in factor selection and data acquisition made the explanatory power of climate factors in this study somewhat subjective.

### 4.2. Application Value

The rapid economic and social development from 2000 to 2020 has led to the expansion of construction land, such as urban and rural settlements, to the periphery of the B-T-H urban agglomerations. Simultaneously, the proportion of living space has increased from 8.3% to 13%, and the production space has decreased from 50.8% to 46.1%, but the proportion of ecological space has not changed much (Figure 6). Ecological space is an important carbon sink and climate reservoir. Because the "Ecological space" is dominated by land types such as grassland and woodland, its carbon sink capacity usually becomes stronger over time [52,53]. From a practical point of view, strengthening the protection and management of ecological space and improving the structure of existing vegetation and the quality of tree stands further improve carbon sequestration [54]. Subsequently, the "Three Zone Space" should be rationally laid out in a mutually beneficial symbiotic relationship to promote the enhancement of carbon sinks.

For future applications, on the one hand, the establishment of an ecosystem-based carbon cycling model is necessary to improve the prediction of the stability of ecological spatial carbon sinks [55]. We suggest that it is necessary to conduct intensive and long-term ecosystem carbon monitoring on a wide geographical scale [56], incorporating meteorological indicators such as wind speed, evaporation, sunshine hours, and ground temperature, in addition to basic precipitation and temperature, to improve the spatial carbon sink simulation model across ecological spaces [57]. With this, we can accurately assess the ecosystem carbon sink capacity and their dynamics under various climate and policy scenarios and regulate climate policies more accurately using simulation results [58]. On the other hand, although climate factors are hard to control, we cannot just sit back and wait for the significant climate impacts; more importantly, we need to find targeted adaptation strategies, such as adjusting the spatial zoning and optimizing land use layout based on the spatial autocorrelation of carbon sinks [59] For example, the design of urban breezeway is based on the direct and indirect effects of wind speed [60], especially for the Beijing–Tianjin Heat Island Cluster; the impact of sunshine hours is used to improve the

plant community and the reasonable layout of green belts [61]. In this paper, the spatially accurate estimation of carbon sinks across ecological spaces under various climate factors can help the government to quickly and accurately adjust the climate policy and realize the visualization of carbon sinks under different scenarios, which is of great significance for China to effectively and timely achieve the double carbon goal.

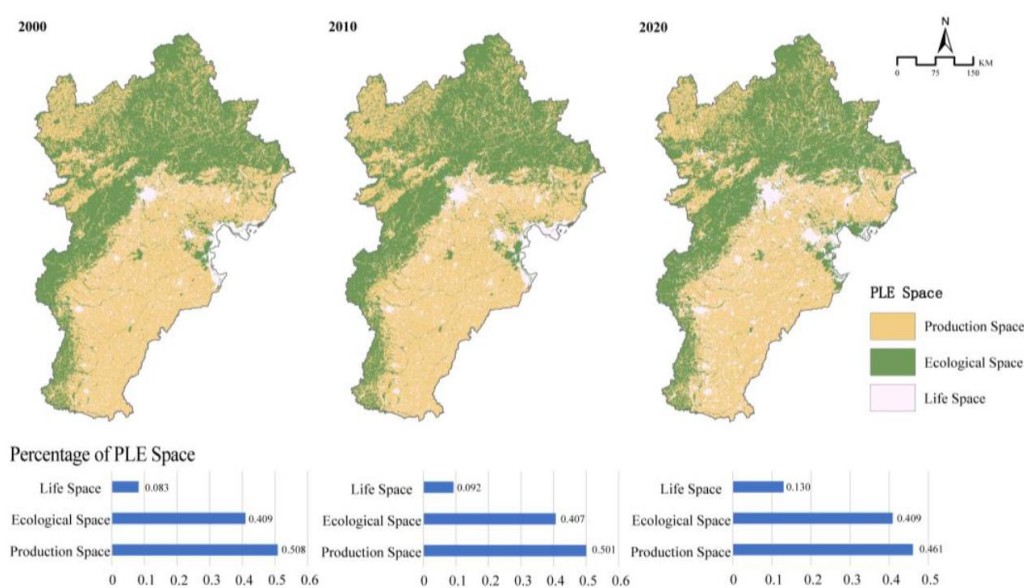

**Figure 6.** Spatial distribution and scale of "Three Zone Space".

*4.3. Future Research*

We can improve the NPP estimation model through the mechanism of climate factors on carbon sinks. On the one hand, the commonly used simulation models, such as plus (patch-generating simulation) [62] and flus (Future Land Use Simulation) models, can benefit from this research and be further developed [63]. By simulating land use development trends and using Invest (https://naturalcapitalproject.stanford.edu/software/invest) (accessed on 1 May 2023) software, it is possible to predict future carbon sinks and the evolution of land use patterns over time; however, additional pre-processing is necessary to improve the accuracy of these predictions by considering the combined effects of $CO_2$ and climate factors. On the other hand, various machine learning-based prediction models, including gray prediction model, power function model, accumulation expansion method, etc. are based on the principle of using the data of biomass density, soil, etc. to calculate the increment of carbon sink for a specific region, which is easy to ignore the spatial effect of the NPP [64]. The calculations should be made from a larger scale, so that the spatial overflow effect can be included in the consideration and the prediction accuracy can be increased. Furthermore, the more established biome-bgc models, in turn, tend to focus only on precipitation and temperature elements; it is common to use only mtclim to make rough calculations [65], and the results obtained are not sufficiently precise. It is better to use observation data for climatic factors with significant spatial characteristics. In future studies, when estimating NPP, especially in urban clusters, it is essential to consider the impact of multiple climate factors simultaneously.

## 5. Conclusions

In this study, we analyzed the spatial effects of different climate-influencing factors on carbon sinks in the ecological space in the B-T-H region using NPP data. Additionally, by using spatial econometric models, the specific climate factors influencing carbon sequestration were identified. We then quantitatively analyzed the contributions of different interannual climate factors to the spatial patterns of carbon sequestration within ecological spaces and spatial spillover effects. The results indicate the following:

(1) A significant increasing trend in NPP from 2000 to 2020 in ecological regions of the B-T-H area, with clear regional and agglomeration characteristics. Therefore, it is advised to proactively adjust land use and climate policies to promote sustainable NPP growth. When implementing ecological spatial governance, it is crucial to consider the interactive effects of urban agglomeration and the surrounding areas and strengthen cooperative governance between regions;

(2) Spatial regression models, especially the SDM, showed the best fit for this study, indicating the presence of both endogenous and exogenous interactions in the model. Under the fixed effect condition, annual precipitation did not have a significant impact on spatial-scale carbon sequestration in the ecological spaces. However, interannual temperature, evaporation, and relative humidity showed a positive effect on carbon sequestration. Annual sunshine duration, ground temperature, and wind speed had a negative impact on carbon sequestration. Considering the spatial autocorrelation of NPP, when building carbon sequestration simulation models, a larger geographic region should be considered, including both the target area and its neighboring spaces, to improve the accuracy of the models;

(3) Various climate factors not only have direct effects on carbon sequestration patterns locally, but they also exhibit spatial spillover effects. Annual average temperature, evaporation, and relative humidity enhanced carbon sequestration locally but reduced regionally. In contrast, annual sunshine duration and ground temperature negatively affected carbon sequestration locally but enhanced regionally. Annual wind speed had a negative effect on NPP both at local and regional scales. Considering the direct and indirect effects of specific climate factors, reasonable recommendations can be made for urban agglomeration and addressing climate change, such as improvement in land use structures and solar and wind resources development.

**Author Contributions:** Conceptualization, X.W. and K.W.; methodology, Y.Z.; software, X.W.; validation, Y.X. and J.G.; formal analysis, X.W.; investigation, Y.Z.; resources, Y.Z.; data curation, Y.X.; writing—original draft preparation, X.W.; writing—review and editing, X.W.; visualization, Y.X.; supervision, Y.Z.; project administration, K.W.; funding acquisition, Y.Z. All authors have read and agreed to the published version of the manuscript.

**Funding:** This research was funded by the Beijing Municipal Education Commission through the Innovative Transdisciplinary Program "Ecological Environment of Urban and Rural Human Settlements", grant number GJJXK210105.

**Data Availability Statement:** The data of the current research are available from the corresponding author on request.

**Conflicts of Interest:** The authors declare no conflict of interest.

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
