# Peer review of "Impact of Climate on the Carbon Sink Capacity of Ecological Spaces: A Case Study from the Beijing–Tianjin–Hebei Urban Agglomeration"

_land, doi:10.3390/land12081619_

Round 1

Reviewer 1 Report

The paper presents how the mechanisms of climate impacts on carbon sinks can contribute to the achievement of carbon neutrality. I would have a couple of suggestions:

- the images are not very clear and connected to the text. Therefore, I recommend increasing the description in their respective references, explaining everything that is depicted, such as colors and legends (for example, in Figure 1, explain why that type of coloring was used and refer to the provided legend if relevant for the article, or in Figure 4, the purpose of the bar graph is not specified);

- in Tables 2 and 6, the meaning of "*" is not specified, while in Tables 4 and 5 it is mentioned at the bottom but in a very small font, so please improve their respective descriptions.

Author Response

  •  the images are not very clear and connected to the text. Therefore, I recommend increasing the description in their respective references, explaining everything that is depicted, such as colors and legends (for example, in Figure 1, explain why that type of coloring was used and refer to the provided legend if relevant for the article, or in Figure 4, the purpose of the bar graph is not specified);
  • Sincerely thank you for your advice. I have redrawn the illustration and improved the description. A legend description has been added to figure 1; a description has been added to figure 4.
  • - in Tables 2 and 6, the meaning of "*" is not specified, while in Tables 4 and 5 it is mentioned at the bottom but in a very small font, so please improve their respective descriptions.
  • Very useful suggestion. I have improved the descriptions

Reviewer 2 Report

I enclose my comments, proposals and corrections.

Reviewer 3 Report

Comments and Suggestions for Authors

The authors at the beginning and at the end of the article reasonably argue for the reduction of active atmospheric gases in China. First of all, the authors look to the future development of society. They have done a verified methodological work, which enhances the effect of reading the article.

General concept comments

Article: Section 3. Results needs to be improved as a more complete report on the results and models is missing.

Review: The conducted studies are disclosed at a high scientific level.

To improve the article, you need to focus on the comments.

Literary references correspond to the context of the article.

Specific comments

1.     Lines 244 and 294. Subsections 3.2.1 Ordinary Least Squares Regression and Selection Tests for Spatial Models, 3.2.2 Spatial Durbin Model Results, 3.2.3 Analysis of Direct and Spatial Spillover Effects, and 3.2.4 Robustness Test, statistical and mathematical analysis are not sufficiently disclosed. The authors in a very short form describe the obtained values of the tables. You need to increase your focus on causality and provide a broader conclusion to your data.

Rating the Manuscript

  • Novelty: The authors touch upon the topical issue of strengthening climate change for the further development of society and minimizing the consequences of an accelerated increase in temperature.
  • • Scope: Ecology, climate.
  • • Significance: The studies carried out are in demand for the rational use of forests, urbanized areas and strengthening the policy of environmental protection. To develop acceptable models for reducing greenhouse gas emissions by interested countries.
  • • Quality: In general, the work was done at a high scientific level and requires minor changes.
  • • Scientific Soundness: There is no doubt about the need for this work. The interrelationships of the biosphere with NEP are given in sufficient detail. About the role of one or another factor influencing the variability of changes occurring in NEP.
  • • Interest to the Readers: The article complies with the policy of the journal. It will be useful to interested persons, scientists and persons involved in ecology and forest protection. Researchers developing climate models and predictive estimates of changes in the structure of greenhouse gases and their impact on the ecosystem.
  • • Overall Merit: The article deserves to be published.
  • • English Level: The article is written clearly.

Accept after Minor Revisions: The paper can in principle be accepted after revision based on the reviewer’s comments.

Author Response

  • Lines 244 and 294. Subsections 3.2.1 Ordinary Least Squares Regression and Selection Tests for Spatial Models, 3.2.2 Spatial Durbin Model Results, 3.2.3 Analysis of Direct and Spatial Spillover Effects, and 3.2.4 Robustness Test, statistical and mathematical analysis are not sufficiently disclosed. The authors in a very short form describe the obtained values of the tables. You need to increase your focus on causality and provide a broader conclusion to your data.
    Sincerely thank you for your suggestions, the Results section has been further refined based on the suggestions.
